# Ceramics for Microelectromechanical Systems Applications: A Review

**DOI:** 10.3390/mi15101244

**Published:** 2024-10-09

**Authors:** Ehsan Fallah Nia, Ammar Kouki

**Affiliations:** Department of Electrical Engineering, École de Technologie Supérieure (ÉTS), Montreal, QC H3C 1K3, Canada; ammar.kouki@etsmtl.ca

**Keywords:** MEMS, ceramics, additive manufacturing, 3D printing, microfabrication

## Abstract

A comprehensive review of the application of different ceramics for MEMS devices is presented. Main ceramics materials used for MEMS systems and devices including alumina, zirconia, aluminum Nitride, Silicon Nitride, and LTCC are introduced. Conventional and new methods of fabricating each material are explained based on the literature, along with the advantages of the new approaches, mainly additive manufacturing, i.e., 3D-printing technologies. Various manufacturing processes with relevant sub-techniques are detailed and the ones that are more suitable to have an application for MEMS devices are highlighted with their properties. In the main body of this paper, each material with its application for MEMS is categorized and explained. The majority of works are within three main classifications, including the following: (i) using ceramics as a substrate for MEMS devices to be mounted or fabricated on top of it; (ii) ceramics are a part of the materials used for an MEMS device or a monolithic fabrication of MEMS and ceramics; and finally, (iii) using ceramics as packaging solution for MEMS devices. We elaborate on how ceramics may be superior substitutes over other materials when delicate MEMS-based systems need to be assembled or packaged by a simpler fabrication process as well as their advantages when they need to operate in harsh environments.

## 1. Introduction

Over the past three decades, tremendous research has been conducted on microelectromechanical systems (MEMS) technology and its potential. MEMS devices, be they sensors, switches, microfluidics, optics, and more, have been extensively studied and applied in different areas of application. Cellular and other wireless connections, medical devices, defense, and space are some prominent areas in which MEMS technology is utilized [1,2,3,4,5,6,7,8,9,10,11]. Fabrication of MEMS devices has always carried many challenges and considerations. MEMS devices are usually fabricated using surface microfabrication or bulk fabrication on a polished substrate with a surface roughness around tens of nanometers. These substrates are usually selected depending on the required application and can be categorized as silicon, glass, ceramic, and metals [12,13,14,15]. They may be used as a functional or non-functional substrate. Over the past two decades, extensive work has been conducted on the fabrication of MEMS devices on non-functional substrates. On the other hand, to benefit from all the potentials of the MEMS devices, they need to be integrated/mounted with other circuits and sub-systems. To realize this goal, functional substrates from different materials and fabrication technologies are developed. Depending on the required application, MEMS devices, after or during fabrication, may be integrated or packaged with a suitable process [16,17,18,19]. Most devices for integration or packaging go through the Complementary Metal-Oxide-Semiconductor (CMOS)-MEMS process. In this process, MEMS devices may be integrated with CMOS ICs as active or passive components. This process is mainly categorized in different approaches: (1) MEMS devices and CMOS circuits are fabricated in different processes and then are integrated or packaged together by different techniques like bond wiring, flip-chip, substrate transfer, etc. The main advantage of this type of integration is that different processes related to MEMS and CMOS do not interfere with each other. However, the final integration as a post-process fabrication may face costly and challenging processes, subsequently affecting the final performance of the whole integrated circuit [20,21]. (2) Integration is conducted in a monolithic process and MEMS and CMOS are fabricated together [22,23,24,25]. While this process eliminates the drawbacks of the first approach in terms of degradation of the final devices and creates a more compact and miniaturized device, it still suffers from interference in the fabrication process with different materials. High temperatures during the CMOS process may introduce mechanical and thermal stress and deformation to materials used in the MEMS process [26,27,28,29,30]. Considering some disadvantages related to the mentioned process in MEMS and silicon-based integrations, other works have been reported with different materials and processes for packaging and integration.

As it is essential for MEMS devices to be stable and reliable in harsh environments, special packaging and integration are necessary for them to tolerate mechanical, chemical, and thermal stresses and corrosion. There are different types of packaging and integration with materials including metal [31,32], plastic [33,34,35,36,37,38], glass, and ceramics [39,40,41,42], which MEMS devices may be embedded, fabricated on the surface of, or fabricated monolithically within as 3D-dense structures. Among these materials, ceramics exhibit excellent performance when it comes to harsh environments. Because MEMS devices are so fragile and delicate at high temperatures, in dusty environments, and from corrosive chemicals, different types of ceramics microfabrication as a standalone microdevice or a host for another MEMS device are suitable options. Compared to other materials like metals and plastics, ceramics have some advantages that make them a better option. As reported in [31], for metals, when it comes to hermetic packaging and the bonding process between MEMS and a metal lid or substrate, a mismatch in the Coefficient of Thermal Expansion (CTE) between silicon-based MEMS devices and metal introduces challenges in bonding through vertical vias. While CTE for metals is in the order of 10 to 20 ppm/°C, different ceramics have lower ranges around 2 to 9 ppm/°C, which is closer to silicon that is 2.6 ppm/°C. There are several works in the literature that illustrate the integration performance of ceramic materials in different conditions [43,44,45,46]. Depending on the availability of the fabrication process and requirements, a variety of ceramic substrates and packages are exploited for MEMS applications. Some of the more prominent materials like low-temperature co-fired ceramics (LTCCs), alumina, zirconia, aluminum nitride, and silicon nitride are reported in several works as ceramic substrates and packages. Ceramic substrates are usually compounds of ceramic, glass, and other chemical solvents with different manufacturing processes. In this review paper, five ceramic materials are chosen for review. The selection of these materials is based on their suitability and unique properties in handling high temperature, electrical and mechanical shocks, and easier integration with MEMS devices, and subsequently, other ceramic materials that are less applicable to MEMS are not included in this review paper. In the second section of this paper, these materials are introduced, and their properties are presented in a table. In the third section of this paper, additive manufacturing (AM) technologies and their sub-section techniques are presented, and among them, we focus more on those ceramics that have more applications for MEMS. Indeed, conventional and other deposition methods of fabricating ceramics are included in section three. In section four, each ceramic, based on its application for MEMS devices and circuits, is individually detailed. In most of the works, three different approaches are governed: (i) ceramics as a substrate for MEMS devices to be mounted or fabricated on top of it; (ii) ceramics are used as a part of MEMS devices or fabricated monolithically in a single process with the same process of MEMS fabrication; and (iii) ceramics as a packaging solution for an MEMS-based system or device. All these approaches are also categorized on a table at the end with their examples that are available in the literature.

## 2. Ceramic Materials and Their Properties

Ceramic materials are widely used in industry and are available in the literature. They are fabricated with different technologies and approaches. The main features of devices fabricated fully from ceramics or partly as an essential unit of a device, including electrical, thermal, and mechanical properties, are explained in detail in different works. Their integration with MEMS and other electronic devices may enhance their performance in different conditions. In terms of electrical properties, high breakdown voltages make the fabricated device more reliable with electrical shocks and possible short circuits in any embedded or integrated circuit. In thermal properties, the low CTE of many of the ceramics lets them integrate more easily with silicon-based MEMS devices like sensors and switches. Also, high thermal conductivity enables them to distribute temperature in the whole circuit uniformly with less energy waste and higher performance. In aspect of mechanical properties, ceramic materials exhibit high robustness when confronting mechanical shocks, cracks, and corrosion. Many MEMS devices including micro-heaters, high-temperature sensors, and RF MEMS devices, be they switches, filters, and phase shifters for arrays of antennas, are able to operate with higher performance and lower energy consumption in harsh environments [47,48,49,50,51,52]. There are a variety of ceramics that are manufactured in industry. Among them, Alumina, Zirconia, AlN, Si_3_N_4,_ and LTCC, which are the focus of this review paper, are chosen (Table 1). These five materials are selected because they are more widely used in the literature, particularly in MEMS technology. Based on this body of literature, these materials showed more suitability for integration with MEMS devices due to their properties. In the following section, each material, in terms of manufacturing, its application and integration with MEMS, and related challenges for them, are explained.

## 3. Ceramic Manufacturing Technologies

Ceramics are manufactured with different approaches, and each has its own benefits and challenges. Each fabrication method affects the properties of ceramics. Conventional methods of fabrication of ceramics are injection molding, dry pressing, sol-gel casting, tape casting, and iso-static press. In Figure 1, we can see the flowchart of ceramics formation from start to final product [55]. In the first step, a specific powder, based on the final product, for instance, alumina, is selected, then the powder is mixed with a specific binder in dry form or with certain slurries. In the third step, the mixture is formed into a desired shape which is called green body. Following this step, a pre-consolidation step called brown body is performed. In the fifth step, a sintering process is conducted in a furnace at elevated temperatures (from 800 °C to more than 1000 °C depending on the ceramics type). Shrinkage happens to the ceramics in this step, which is important in the case of MEMS applications and should be controlled and compensated when alignment is needed with MEMS devices. Indeed, in the final step, machining is conducted for the required dimension and surface finish. Cutting, laser ablation, and polishing are forms of machining. For MEMS applications, polishing is critical to reach a very smooth surface in the range of tens of nanometers. For more advanced technologies and approaches in ceramics manufacturing, AM, also known as 3D-printing technology, is widely reported in the literature. As reported in [56,57], based on the ISO standard classification of AM technologies, there are seven main categories with sub-techniques in parentheses: (1) Vat photopolymerization (Stereolithography Apparatus (SLA), Digital Light Processing (DLP), Liquid Crystal Display (LCD)), (2) sheet lamination (laminated object manufacturing (LOM), ultrasonic additive manufacturing (UAM)), (3) Powder bed fusion (Selective laser sintering (SLS), Selective laser melting (SLM), electron beam melting (EBM), direct metal laser sintering (DMLS)), (4) directed energy deposition (laser-based metal deposition (LBMD), electron beam freeform fabrication (EBF3)), (5) material jetting (continuous stream mode (CS), drop on demand mode (DOD)), (6) binder jetting (BJ), and (7) material extrusion (fused filament fabrication (FFF), fused deposition modeling (FDM), Direct Ink Writing (DIW)). Each of these technologies has different techniques and methods which are suitable for certain applications (Figure 2). In recent years, the disadvantages of conventional methods like shrinkage, time-consuming sintering process, and demanding complex ceramic structures and rapid manufacturing, caused the emergence of AM technologies. complicated shapes like light-weighted structures in biomedical, aerospace, and automative applications are realizable through AM technologies [58]. On the other hand, it is widely reported that the very high melting point of ceramics is among the challenges when using AM technologies and layer-by-layer manufacturing of ceramics. Due to these challenges, some post-processing technologies are reported in [59,60,61,62]. In the following sections, the techniques that are most used in manufacturing the five mentioned ceramics and their suitability to be applied in MEMS technology are detailed. In Figure 3, all these methods are illustrated [63]. Indeed, in Figure 4, it is shown that these processes may be manufactured to produce single-material products or multi-material ones based on the utilization of a combination of materials in a hybrid manufacturing process [64].

### 3.1. Additive Manufacturing Technologies

In the following section, AM technologies and their sub-techniques that are used most in the literature for five ceramics and their suitability for MEMS applications are presented.

#### 3.1.1. Vat Polymerization

In this technique, which is based on a source of light, the light aims at a photopolymer material and makes changes to its properties, like hardening it. Two techniques, including SLA and DLP, are used to manufacture different types of ceramics. In some of the recent advancements in AM of alumina fabrication, for example in [65], the author used Digital Light Processing (DLP) technology to fabricate alumina/calcium phosphate samples with microporous features to apply in lightweight and high-aspect-ratio structures. In this work, a high-quality and lightweight alumina ceramic is produced while the cost of fabrication is kept low. In another work [66], authors introduce a new way of using AM to produce alumina by using a layer-by-layer printing process. Compared to the reference in this paper illustrated to be 650 MPa, they achieved a very high biaxial strength of 1 GPa in fabricating alumina ceramic. In this process, a multi-material approach is used. Layers of alumina and zirconia ceramics are sandwiched together to form a stronger material compared to the fabricated monolithic alumina in another work. In Figure 5, two different fabricated ceramics are shown. Stereolithography by applying 3D-printing technology is used to fabricate alumina [67]. As the sintering process is difficult in alumina due to the high melting point, some aiding particles such as TiO_2_, CaCO_3_, and MgO are added to the sintered material. As a result, the flexural strength of fabricated ceramic is increased from 139.2 MPa to 216.7 MPa. Also, the anisotropic shrinkage of the final product is decreased, as shown in Figure 6. Like alumina preparation, SLA is used for zirconia in [68]. For Zirconia, parameters like raw materials, slurry, debinding, and sintering in these processes all affect the final product. Aluminum Nitride plays an important role in many applications and is well-reported. Optoelectronics, MEMS, and packaging electronic devices, especially for wireless communication (e.g., piezoelectric materials) are other important areas where AlN is used. As mentioned in earlier sections, AM technologies are applied to enhance the properties of AlN ceramics. For instance, in [69], Digital Light Processing is used to fabricate AlN with high thermal conductivity and density. In this process, the pressure of nitrogen gas and sintering temperature in the process are key factors to produce AlN with high quality. The optimum sintering temperature is determined to be 1720 °C while nitrogen gas pressure is kept below 1 MPa (around 0.6 MPa). The final AlN ceramic has a thermal conductivity of 168 W·m^−1^·K^−1^ and a density of 3.35 g/cm^3^. A simple process is illustrated in Figure 7.

#### 3.1.2. Powder Bed Fusion

In powder bed fusion, a source of laser melts ceramic powders and forms a solid material. In [70], the author applied laser powder bed fusion to fabricate alumina ceramics with a very low shrinkage rate with adequate strength, which may be suitable for MEMS that need precise alignment. More efforts are also reported in [71,72], in which the laser-assisted selective and selective laser melting fabrication processes are shown. In these works, a laser is used to form deposited metallization on an alumina substrate to create transmission lines. The first process helps to fabricate the desired metallization on low-loss ceramic substrates like alumina, while the second process helps to fabricate 3D metallization with good adhesion of metal to alumina after heating during the fabrication. Like alumina in the previous section, there are plenty of AM technologies to fabricate zirconia ceramics [73,74,75]. Two important challenges are considered in several works for the manufacturing of zirconia in SLS technology [76,77,78]. First, powder characteristics. The density of the powder is directly proportional to the size of the powder bed and the final structure of the zirconia ceramic, and it could be increased by certain slurry deposition. Indeed, blending the powder with another low melting point solvent may enhance liquid phase sintering and subsequently result in a denser ceramic structure in laser processing. The second parameter is laser parameters. The velocity and power of the laser affect the final manufactured ceramic. As seen in Figure 8, laser power and velocity can clearly determine the surface of zirconia ceramic. For MEMS fabrication on a zirconia sample, a denser compound and fewer cracks and holes may help better surface microfabrication that has a direct influence on MEMS device performance. Another technology for zirconia ceramic fabrication is called laser melting and is almost identical to the previous technology, with one exception. This is a one-step process without any post-process like a low melting step. As reported in other works, the disadvantage of this process is the low density of the final ceramic product. To address this issue, a preheating process is supposed to be the most effective way [76]. Illustrations in Figure 9 show cracks that are more distributed and decreased when the preheated temperature increases in treating ceramic powder. Moreover, in [79], a comprehensive study on defects in ceramics is presented. Challenges and solutions to control defects are extensively studied. Pore, crack, delamination, and surface defects are addressed in this review work.

#### 3.1.3. Binder and Material Jetting

In material jetting technology, the repeated processes of curing different materials with UV light are performed to create a 3D structure. The advantage of this process over Vat polymerization is that no further post-cure process is needed. Moreover, in binder jetting, layers of ceramic powder are attached to each other with a liquid. Inkjet printing is one of the techniques used in binder jetting that is applied in zirconia fabrication. This is a fast and economic process that makes it one of the most suitable approaches in AM of zirconia ceramics. The process starts with the drawing of the CAD file for 3D printing, then the preparation of the ink, printing, and sintering process. To improve LTCC ceramics properties and accelerate the fabrication process while avoiding complexity, AM, like other ceramic materials, is used in the manufacturing of LTCC. Comparing different methods, as reported in [80], one of the best technologies to produce LTCC ceramic substrates is material jetting. In this work, LTCC ceramic ink materials and silver inks are jetted by a 3D printer equipped with a piezoelectric nuzzle print. Matching between LTCC material ink solid particles and silver ink is critical in this process and is selected in a way to have a similar shrinkage rate between ceramic and silver after the sintering process. Indeed, the interaction between ceramic and silver is carefully examined to avoid any significant diffusion of silver into the ceramic (Figure 10 and Figure 11). Since LTCC ceramic is a suitable candidate to be a substrate or package for silicon-based devices, the ceramic material is chosen in a way to have a close value of CTE to silicon. In this work, this value is close to 4.1 ppm/°C. The final product, which is a metallized LTCC substrate is validated through different simple applications, a flat substrate with metallization and a curve substrate by a curved shape-printing process. A microstrip antenna is fabricated and S-parameters are measured between 9 and 11 GHz. Results are in good correlation with the simulation and gain of the antenna, showing good dielectric properties of the fabricated antenna (Figure 12). In the next application, an arc-shaped substrate with metallization is fabricated. The electrical properties and the shape of the LTCC ceramic substrate remained similar to a flat one, even after the sintering process exhibited a successful fabrication (Figure 13).

### 3.2. Non-Additive Manufacturing Technologies

Apart from the ISO classification of ceramic-manufacturing techniques, there are other methods, including conventional and some specific deposition methods in the literature. In the next sections, these techniques are detailed.

#### 3.2.1. Conventional Manufacturing

Conventional fabrication of ceramics commonly follows the same process (Figure 1). For instance, green sheets of LTCC that are used to fabricate substrates and packages for electronic and microwave circuits are widely produced through conventional methods. The combination of different types of ceramics and glass powders with certain slurries as solvents is used to prepare sheets. In [81], depicted in Figure 14, a ceramic material that is used (Cordierite) is prepared by ball milling after calcination at 1350 °C. On the other hand, glass powder is prepared by melting process and then the quenching method. Ceramic and glass are then mixed in a ratio of 30:70 in an IPA solution and dried under an IR lamp. At the end of this step, LTCC powder is prepared. In the next step, to prepare LTCC slurry for tape casting, LTCC powder, as shown in Figure 15, is mixed with certain solvents, then, plasticizers and homogenizers with proper binders undergo ball milling for a full 24 h to prepare the final slurry. The prepared slurry is then processed in a tape-casting method, covers a suitable substrate as mylar, and is dried up to prepare the final LTCC green sheets.

In another work with tape-casting technology, AlN is produced [82]. During the process, Aluminum dihydrogen phosphate is used to prevent AlN surface particles from hydrolysis. The final product is a well-dense structure with a homogeneous distribution of particles across the ceramic. Flexural strength is 283 MPa and thermal conductivity is around 116 W·m^−1^·K^−1^.

#### 3.2.2. Manufacturing through Deposition Techniques

In addition to the conventional and AM fabrication of AlN, there are other deposition methods used in semiconductor and MEMS fabrication of AlN, Silicon Nitride, alumina, and zirconia. These processes usually take place in cleanrooms where MEMS devices are fabricated. Physical Vapor Deposition (PVD) is a way of creating tens and hundreds of nanometer-thick layers of AlN. DC-reactive sputtering is a kind of PVD that is used to create piezo AlN layers [83,84]. Also, chemical vapor deposition is another method and atomic layer deposition is an example of this deposition technique [85]. These processes also apply to Si_3_N_4_ ceramics. PVD deposition of Silicon Nitride is reported in [86], as well as CVD in [87,88]. Alumina e-beam PVD is also reported in [89]. Yttria-stabilized zirconia coatings by PVD are reported to create a dense layer in [90]. In the following section, where the application of ceramics in MEMS devices is studied, more details for final fabricated devices based on mentioned depositions are explained.

## 4. Use of Ceramics in MEMS-Based Systems

The use of ceramics in MEMS-based devices and systems can broadly be classified under three main areas: substrates, monolithic fabrication, and packaging. As non-functional substrates, ceramics, once properly polished, can be used as an alternative to other conventional substrates (silicon, glass, quartz) for MEMS surface microfabrication. In monolithic fabrication, ceramics can be part of an MEMS device (e.g., thin-sputtered layers) or as functional polished substrates on which MEMS devices can be fabricated in a single process. Ceramics can also be used for packaging purposes to cap and seal MEMS devices like sensors and switches. As highlighted before, the ceramics materials that are most frequently used for these applications are alumina, zirconia, AlN, silicon nitride, and LTCC. In the following, we review various applications of each of these materials in MEMS-based systems and provide a comparative summary table (as Table 2) between them at the end of this section.

### 4.1. Alumina-Based MEMS Devices and Systems

In the literature, several works are conducted in the MEMS area using alumina ceramics. As a smooth substrate for surface microfabrication, there are many works that use the potential of alumina, like in high-temperature sensors. In [91], they use alumina as a substrate to fabricate a Pt sensor. A fabricated sensor is able to work at elevated temperatures (up to 900 °C) due to the very high melting temperature of alumina (Figure 16). In [92], the metallization of the alumina substrate by Cr, Cu, NiP (Nickel-plated), and Au is performed for the purpose of microwave-integrated circuits. Due to the very low dielectric loss of the alumina substrate, microwave circuits can operate without significant insertion loss. In [93], authors demonstrated an ultra-miniaturized microwave circuit using 80 µm thick alumina ribbon as the substrate. Band Pass Filters and Low Pass Filters are fabricated and the top and bottom of the circuit are connected with internal vias through alumina ribbon. Circuit performance at 28 GHz is compared to the other conventional circuits available in the literature and results are promising. Ultra-miniaturized alumina ribbons help fabricate very small MEMS devices like microwave filters and sensors without losing performance. In [94], a Ka-band cavity filter is fabricated with alumina as a dielectric. In this circuit, alumina is produced with AM technology as 3D printing. This design let the first fourth-order prototype with a high-quality factor (800). The return loss of the cavity filter is 13 dB and the insertion loss is 1.3 dB. Narrowing the size of the irises led to this loss. To avoid this problem, another fabrication of alumina with 3D-printing technology is performed, but this time with a higher resolution. Final devices have the potential to be used as a surface mounting unit due to the compact design (14 × 25 mm^2^).

In another type of application, alumina can be a part of MEMS device structures instead of being a substrate for another MEMS device. In [95], an optical miniaturized device as an anti-reflective structure is fabricated using alumina ceramic. First, an aluminum substrate is machined to form a concave structure. After that, a two-step process of anodic oxidation of alumina is conducted on top of that to create nanopores and arrays of nanonipples. The performance of the fabricated lens in terms of light transmission is illustrated compared to different sizes of pores and nipples and a normal lens which reflects the enhancement of the fabricated device (Figure 17). In another work [96], a similar process for creating alumina is conducted. Anodic oxidation of aluminum created an alumina porous layer with an integrated ignition heater. The alumina layer is a support layer for the Pt catalyst. The measurement shows a heat release rate of around 830 MW/m^3^.

In [97] as a sensing purpose, a Relative Humidity (RH) capacitive sensor is fabricated. The sensing layer is a porous alumina layer and the whole process is conducted by a monolithic CMOS-MEMS process. Based on this paper, sensing of the fabricated device is much higher than the average of the literature and is around 15 pF/RH% compared to 0.2–0.5 pF/RH% in other works. In another work for sensing purposes, high-temperature handling of alumina is the main feature of the fabricated sensor. A micro-hotplate is fabricated, which comprises gallium oxide as a sensing layer and an alumina-suspended membrane that is sandwiched between two platinum layers. The alumina layer, compared to silicon nitride, can handle 650 °C for 110 h, while with silicon nitride, the micro-sensor is damaged within a few seconds. Carbone monoxide and nitrogen dioxide were sensed with the fabricated device under high temperature. The fabricated device under test and its performance are shown in Figure 18 [98].

Finally, when it comes to the packaging of the MEMS devices, alumina can be a suitable candidate. There are several works demonstrating the application of alumina. In [99], a wafer-level MEMS fabrication using an alumina nanopore layer is conducted. The alumina layer is around 2 to 3 µm and is sealed with silicon nitride to make the packaging stronger. Pores are 15–20 nm, and they are fabricated with the silicon oxide sacrificial layer and an HF release process. The fabricated RF device as a CPW transmission line is measured with s-parameters and the RF loss due to the package is very negligible up to 67 GHz (Figure 19). Another similar work also has been conducted to encapsulate the MEMS device with an anodic process of alumina. The pressure inside the package after the glow discharge due to the high voltage on the MEMS-suspended membrane is measured and is decreased from 50 µtorr to 2 µtorr [100]. In [101], nanopillars of alumina with deep pores around 20–30 µm are fabricated. the high aspect ratio fabricated alumina pillars can be a good housing for many MEMS devices for packaging purposes under high-temperature environments (Figure 20).

### 4.2. Zirconia-Based MEMS Devices and Systems

Zirconia ceramic materials are used in several works involving MEMS structures, as a substrate, MEMS device, and packaging solutions. As a substrate, in [102] a zirconium oxide membrane is fabricated for sensing purposes. The membrane is fabricated using slip casting under mechanical pressure and annealing process. Particle sizes of fabricated membranes are around 20 nm. The sintering process is also reported to be 1150 °C. The fabricated membrane is about 10 µm thick and the roughness is appropriate for the deposition of sensing the platinum layer. The application of zirconium oxide is due to its low CTE, which is about 2.5 W·m^−1^·K^−1^. This makes the sensor consume less power compared to an alumina membrane. The fabricated sensor is suitable for combustible gases and gas-fire detections. In another study [103], different substrates are used for an MEMS pressure sensor to analyze the effect of each substrate. Different ceramic substrates including AlN, Si_3_N_4_, Zirconia-silicate, and LTCC are used. Among them, LTCC and zirconia-silicate exhibit better performance in terms of mechanical and thermal stress reduction and cross-sensitivity enhancement, which totally increases the reliability of the MEMS device (Figure 21). As a flexible substrate, Yttria-Stabilized Zirconia (YSZ) is used to fabricate thin-film solar cells. The surface of the substrate is smooth, and the roughness (20 nm) does not need any modification like polishing. Copper indium gallium diselenide (CIGS) is deposited on top of the YSZ substrate. A 3000-angstrom molybdenum layer was deposited on the back of the substrate. Then, a 120-angstrom-thick layer of sodium fluoride is evaporated on top. Then on top of that, a 2 µm-thick CIGS layer is evaporated. The fabricated flexible device is illustrated in Figure 22. As reported in this paper, this device may open the door to fabricating many flexible MEMS devices, like high-temperature sensors that are able to work in high temperatures [104].

In [105] a zirconia MEMS-based micro-thruster is fabricated using a gel-casting method on polydimethylsiloxane (PDMS). In Figure 23, process fabrication is presented. After casting the zirconia ceramic suspension on PDMS and machining to form the structure, conductive pastes are printed, and after sealing, the device is sintered. Shrinkage after sintering is around 10–15%. Based on this paper, the formation of the structure can further be enhanced by using other techniques in MEMS fabrication like lithography. Current fabrication is a prototype that can be used for micro-propulsion systems that are able to work under harsh environments such as high-temperature, corrosive, and oxidative. In [106], an ammonia sensor is fabricated using 0.18 µm CMOS technology integrated with a readout circuit. The sensor has interdigitated electrodes and a sensitive film, which is a zirconium oxide layer, and is deposited as a post-processing approach. Post-processing includes a sacrificial layer deposition and etching to create an opening by following the sol-gel method, which involves dropping zirconium oxide by micro-dropper and then calcinating it at 100 °C. The fabricated sensor and measurement results are available in Figure 24; the sensitivity of the sensor when ammonia gas is introduced at about 4.1 mV/ppm is also shown.

For the packaging aspect, titanium, zirconium, and Zr-Ti alloy are deposited under an ultra-high-vacuum condition. Grain sizes of layers are measured, and their values differ when the thickness of a deposition charges. It is found that the single metal layer has a lower grain boundary density. While it is stated in this paper that higher grain boundary density may lead to higher absorption of gases, the alloy has a better absorption, in conclusion. So, for the packaging purposes of an MEMS device, using a Zr-Ti alloy layer with a lower thickness may help to absorb more gases during the activation of an MEMS getter layer [107].

### 4.3. Aluminum Nitride-Based MEMS Devices and Systems

Due to the high thermal conductivity, the decent electrical properties of AlN, and its compatibility with other materials in the fabrication of MEMS devices, like silicon in the CMOS-MEMS process, it has been widely used and reported in the literature for the past decades. Since MEMS fabrication needs a very smooth surface, in [108], AlN is reported as a substrate that needs to be polished before any MEMS fabrication. For this polishing and lapping process, SiO2 is used as a slurry, and other parameters like load on the sample and rotation speed are tuned in a way that the final roughness of the surface is around 6 nm, which is excellent for MEMS fabrication. Indeed, gaps found on the finishing surface of the substrate have an adverse effect on the fabricated MEMS device as well as AlN properties. For this reason, as advised in this paper, the AlN substrate should be as compact as possible before the final sintering of the ceramic (Figure 25). Following this issue in [109], aluminum nitride-yttria ceramics are produced, aiming for a better substrate. Mechanical properties, including flexural, are enhanced through pressure-assisted two-step sintering at 1680 °C. After sintering, a post-process, which is microstructural freezing, is conducted. The result grain size of the ceramic is reduced from 2.21 µm to 1.08 µm.

As an MEMS device, AlN has gained great attraction during the past years, especially as a piezoelectric device. Capacitive Micromachined Ultrasonic Transducers (CMUTs) and Piezoelectric Micromachined Ultrasonic Transducers (PMUTs) are among the most reported devices in the literature [110,111,112,113]. Due to the need for high voltage in CMUTs and small voltage in PMUT devices, the latter attracted more people. MEMS microphones, speakers, and energy harvesters benefit from the piezoelectric feature of AlN. In Figure 26, some of the PMUT devices utilizing AlN and Si_3_N_4_ as ceramic materials are shown (Figure 27) [114,115,116]. In the telecommunication area, AlN MEMS-based devices are widely reported and investigated. In [117,118,119,120], AlN Bulk Acoustic Wave (BAW), Surface Acoustic Wave (SAW), and lamb-wave resonators and filters are among the most-reported devices. in Figure 28, some of the 3D and cross-section views of these devices are shown.

For the packaging of MEMS devices, since AlN has excellent thermal and electrical properties like high thermal conductivity, low CTE, and high insulation, it is a good candidate for packaging MEMS high-power devices as heat sinks. In [121], a novel AlN filler approach was used to produce a high-thermal conductive resin for packaging MEMS devices. The thermal conductivity of epoxy resin filled with AlN filler in this work could reach 12 W·m^−1^·K^−1^.

### 4.4. Silicon Nitride-Based MEMS Devices and Systems

Intense research has been conducted for several years on silicon nitride ceramics when it comes to their application on MEMS. Apart from the preparation of Si_3_N_4_ with different approaches in the AM area, there are other techniques available in the MEMS area for producing Si_3_N_4_ thin films and substrates. Among them, we can mention CVD, PECVD, and ALD, which all lead to a thin film of Si_3_N_4_. As an application in the substrate of an MEMS device, Si_3_N_4_ is usually used to cover silicon wafers. High mechanical strength, gradual stress control, and corrosion prevention of silicon during the wet etching process are reasons for covering silicon wafers with Si_3_N_4_. Almost all MEMS devices can benefit from Si_3_N_4_ properties. In space and communications, biomedical, optics, and automobiles, we can mention RF-MEMS, sensors, and optical MEMS devices that use Si_3_N_4_ as a passivation layer when contact between RF and DC routes happens. On the other hand, it can be used as a dielectric layer to create a parallel plate capacitor. Due to its high dielectric constant and breakdown value, it is possible to fabricate very thin capacitors with high capacitance ratios that can handle high voltages [122,123,124,125]. Advances in silicon photonics in the past three decades, thanks to the properties of Si_3_N_4_, and its compatibility with CMOS technology, opened new doors to enhance optical communications in silicon-based MEMS devices, whether in the passive or active domain. As a passive optical MEMS device, we can mention couplers and splitters reported in [126,127], while as an active device, which is compatible with CMOS technology, we can mention the micro-ring modulator and PZT-covering Si_3_N_4_ waveguide [128]. For switching purposes in several works, Silicon Nitride optical switches are reported. In [129], actuators with induced voltage move the optical waveguide and couple signals between the fixed and movable lines. Loss in this configuration is measured between 4.64 dB and 5.83 dB. As reported in this paper, this device promises improved tunable transceivers operating in C-band. Silicon nitride is also applied to the packaging of sensitive MEMS devices (Figure 29). As an example in [130], a capped BAW MEMS device is illustrated. The sealing of the MEMS device is implemented in a way that handles mechanical damage and loads due to external pressures from the dicing and wire-bonding wafer. As a result of this work, it is reported that this type of packaging can protect MEMS devices during mass production (Figure 30).

### 4.5. LTCC-Based MEMS Devices and Systems

LTCC ceramic materials can be a suitable option for integration with different MEMS devices, including sensors, microfluidics, and switches. Since LTCC ceramics are fabricated as the layer-by-layer process (conventional process) or as a whole 3D structure by printing (AM process), they could be used either as a package, a substrate with embedded passive components, or as a miniaturized device itself. As a substrate, if a conventional method is used to fabricate LTCC, the surface of the substrate before any MEMS process needs to be polished to reduce surface roughness to tens of nanometers. There are several works reporting this issue and solutions to overcome it. Usually, a chemical mechanical polishing machine with a proper slurry that includes nanometer-sized particles is used to reduce roughness as well as waviness of the substrate. As a non-functional substrate, LTCC, like other ceramics mentioned in previous sections, and other conventional substrates, like silicon and glass, are used for surface microfabrication. In [131], RF-MEMS switches are fabricated on top of LTCC substrates. In [132,133], micro-heaters are fabricated either by PVD sputtering after polishing or by metallization by the LTCC process. LTCC substrates, on the other hand, can be fabricated as functional substrates to embed metallization that either could be DC or RF lines, cavities formation, and vertical vias from the bottom to the top of the LTCC. There are several works that report the integration of the functional LTCC with previously fabricated MEMS devices. MEMS chips could be bonded to the LTCC by bonding from the chip contact pads to the LTCC vertical vias. In [134], a novel monolithic LTCC-MEMS process is reported, which is one process, LTCC and MEMS devices are fabricated together without the need for challenging post-processing, like alignment for bonding in other processes like the CMOS-MEMS process. Due to the nature of LTCC, which has a rough surface and shrinkage in X, Y, and Z directions, solutions are offered to facilitate the fabrication process as a monolithic one. The roughness of LTCC surface plus via bumps are controlled in a custom polishing process, and shrinkage is compensated with a straightforward solution to fabricate MEMS on top of LTCC vias with less than few micron misalignments. To prove that the process is practical, a capacitive RF-MEMS device is fabricated on top of the functional LTCC substrate while RF and DC control routes are buried inside LTCC layers. As shown in [134], RF performance is reported, and it is on par with what is available in the literature as a low-loss switch (Figure 31 and Figure 32). LTCC ceramics can be used as a miniaturized device itself. There are several works reporting LTCC devices as microfluidic devices. Like in [135], as a gas flow sensor, holes act as channels. In [136], pressure sensors are reported that are integrated with other electronic chips. Humidity sensors are also reported in [137]. Since LTCC could be fabricated layer-by-layer as a 3D structure, multifunction sensor fabrication is also feasible. Like in [138], LTCC acts as a fluidic device hosting other heaters and optical sensors as a whole multifunction system. A movable capacitive force sensor as a cantilever with a one-side fixed anchor is reported [139]. A special sacrificial layer during the fabrication of an LTCC is used to create the gap between the cantilever bridge and the bottom electrode, which will be removed later for the structure. LTCC is also used for packaging and its implementation is well-reported for different RF-MEMS, MOEMS, sensors, and optical circuits [140]. Some examples are shown in Figure 33, Figure 34 and Figure 35.

**Table 2 micromachines-15-01244-t002:** Application of different ceramics in three different categories for MEMS.

Ceramic	Use in MEMS		
	Substrate	Monolithic/Part of Device	Package
Alumina	Temperature Sensor [91], Microwave Circuit [92,93,94]	Optical MEMS device [95,96], RH sensor [97], gas sensor [98]	Wafer-level packaging [99,100,101]
Zirconia	Gas sensor [102], pressure sensor [103], flexible substrate [104]	Micro-thruster [105], Ammonia sensor [106]	Alloy for gas absorbing in package [107]
AlN	Machining substrate for MEMS [108,109]	CMUT, PMUT [110,111,112,113], SAW, BAW resonators [114,115,116]	Heatsink package [121]
Si_3_N_4_	Passivation layer for Capacitors [122,123,124,125]	Optical coupler, splitter [126,127], Micro-ring modulator [128], Optical Switch [129]	Capping and sealing BAW resonator [130]
LTCC	RF-MEMS switch [131], Micro-heater [132,133]	Monolithic LTCC-MEMS process [134], gas flow sensor [135], pressure sensor [136], humidity sensor [137], fluidic [138], cantilever [139]	RF/Optical Switch, sensors capping and embedding [139]

## 5. Conclusions

In this review paper, the use of ceramics for MEMS devices and systems is discussed. Because MEMS devices are delicate and fragile, there is a need to mount, assemble, and pack them. Furthermore, MEMS devices may need to operate in harsh environments, where they are susceptible to high temperatures or chemical corrosion. For these and other considerations, ceramics can be the materials of choice for MEMS-based systems. The unique properties of ceramics, including close CTE values to silicon and high tolerance to elevated temperatures, mechanical and electrical shock, and their extensive use in other non-MEMS circuits and packaging applications make them suitable for MEMS-based systems. While there is a large variety of ceramic materials, the focus has been placed on the five most frequently used ones in this review, namely, alumina, zirconia, aluminum nitride, silicon nitride, and LTCC. Conventional manufacturing of ceramics has been a common process used for many years. However, due to the emergence of 3D-printing technology, additive manufacturing of ceramics paved the way for faster and easier fabrication of more complex ceramic structures, avoiding the lengthy and costly post-process steps associated with conventional methods. Consequently, more applications using ceramic-based MEMS devices are now available in simpler and more efficient ways. Looking ahead, it is expected that as ceramic-based MEMS devices and systems continue to develop and mature, they will be able to offer more reliability and cost-effectiveness in many systems such as automobiles, factory and plant safety systems, biomedical and communication equipment, and more.

## Figures and Tables

**Figure 1 micromachines-15-01244-f001:**
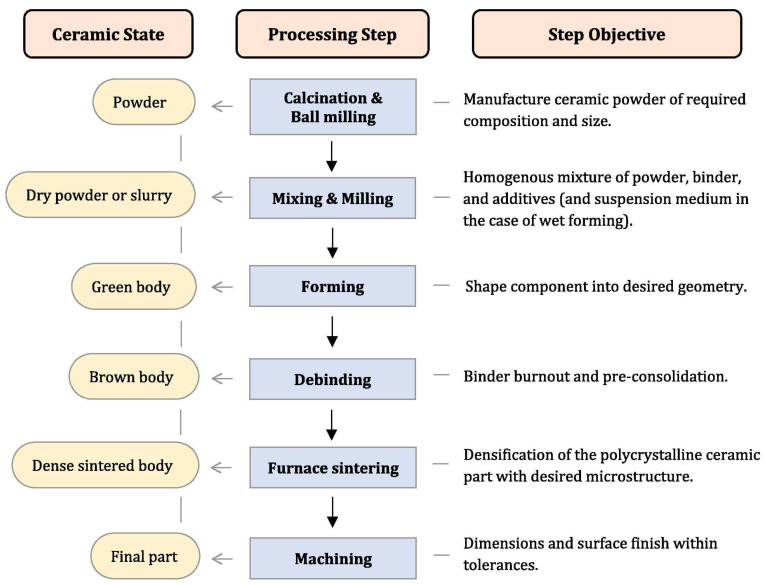
Conventional manufacturing of ceramics from beginning to final product [55].

**Figure 2 micromachines-15-01244-f002:**
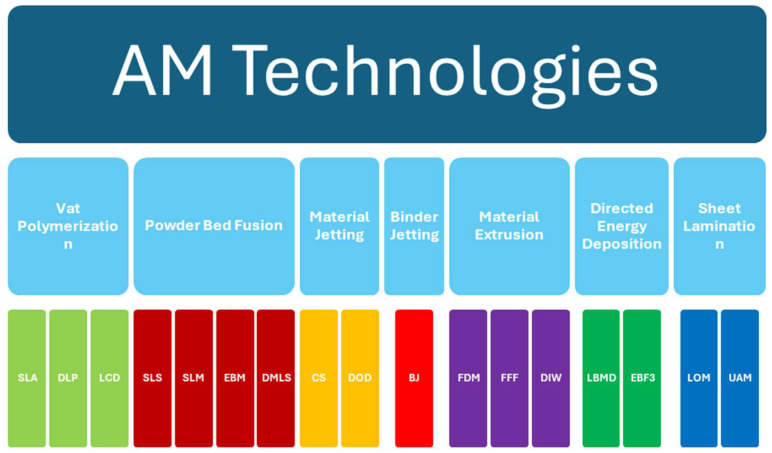
Different types of additive manufacturing and their techniques based on ISO classification [56,57].

**Figure 3 micromachines-15-01244-f003:**
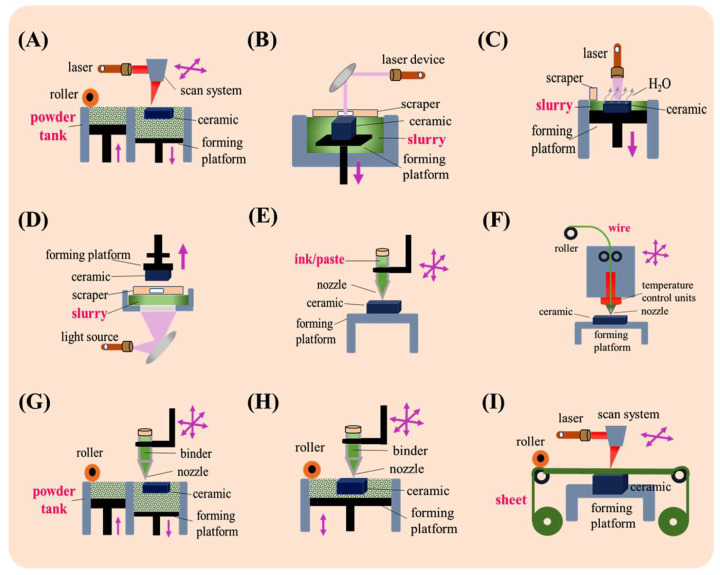
AM approaches for Si_3_N_4_ manufacturing: (**A**) SLS/SLM; (**B**) SLA; (**C**) LIS; (**D**) DLP, LCD; (**E**) DIW; (**F**) FDM; (**G**) BJ; (**H**) 3D printing (3DP); (**I**) LOM [63].

**Figure 4 micromachines-15-01244-f004:**
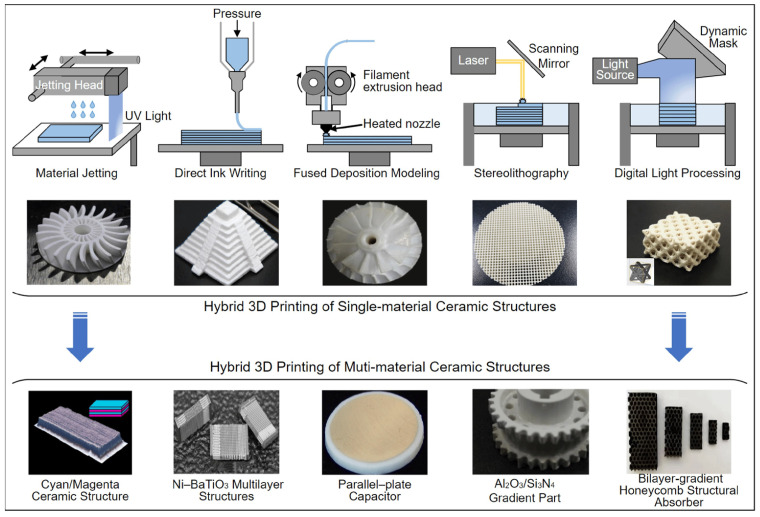
Five different 3D-printing techniques: Digital Light Processing (DLP), material jetting (MJ), Stereolithography (SLA), Fused Deposition Modeling (FDM), Direct Ink Writing (DIW) [64].

**Figure 5 micromachines-15-01244-f005:**
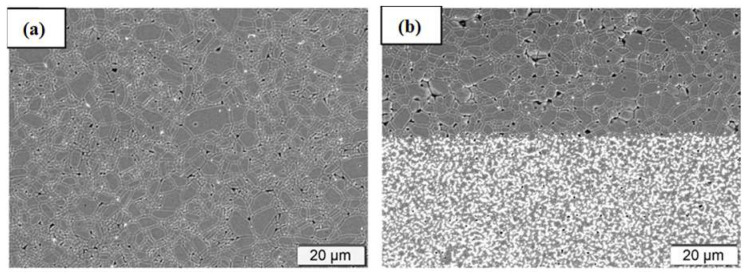
Microstructure of (**a**) monolithic (**b**) multi-structure of fabrication [66].

**Figure 6 micromachines-15-01244-f006:**
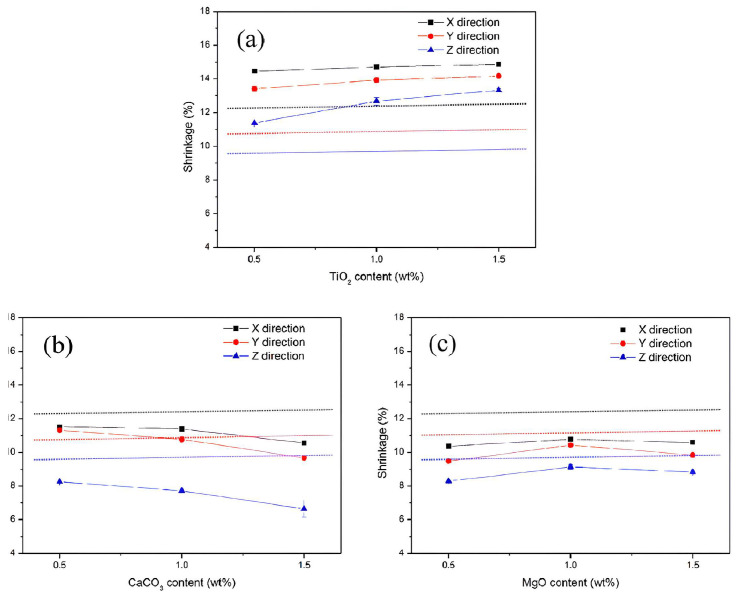
Shrinkage rate of the sintered ceramic with different sintering materials content, (**a**) (TiO_2_) on top, (**b**) (CaCO_3_) on left and (**c**) (MgO) on right in all direction (X direction in black, Y direction in red and Z direction in blue), direct lines indicate the shrinkage before adding materials [67].

**Figure 7 micromachines-15-01244-f007:**
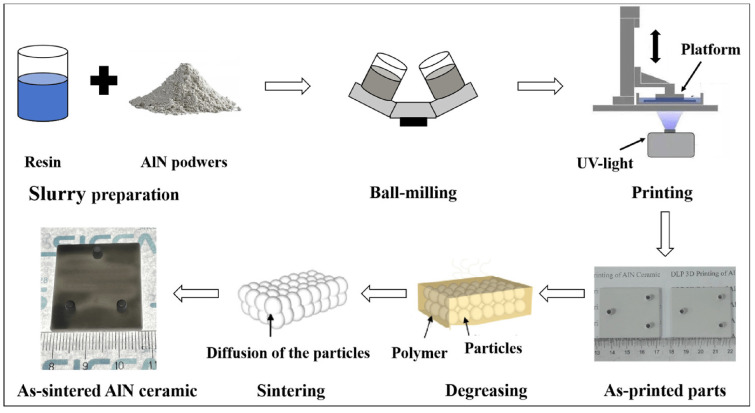
DLP technique for AlN 3D manufacturing [69].

**Figure 8 micromachines-15-01244-f008:**
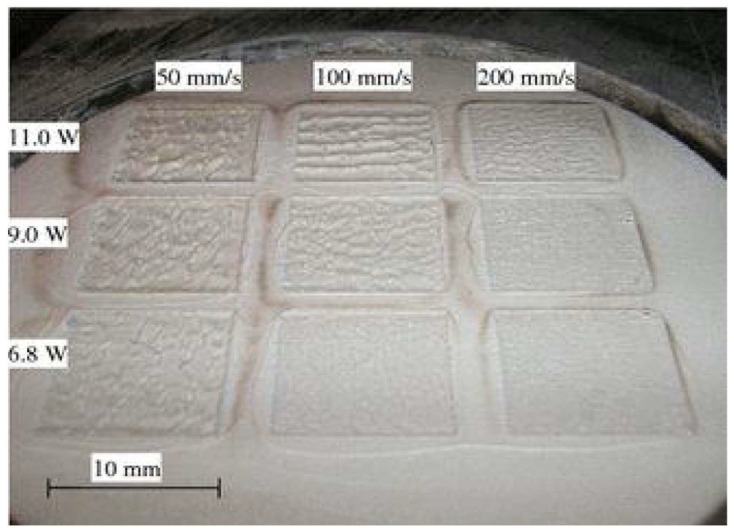
Laser power and velocity variation effect on surface morphology of zirconia sample [76].

**Figure 9 micromachines-15-01244-f009:**
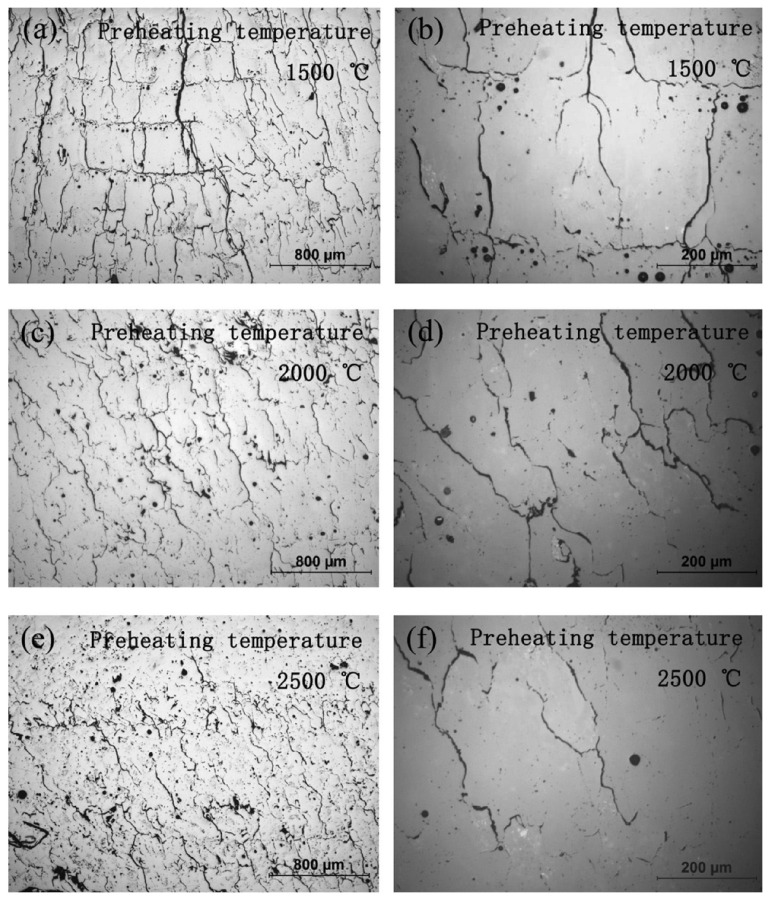
Preheat temperature effect on sample cracks [76].

**Figure 10 micromachines-15-01244-f010:**
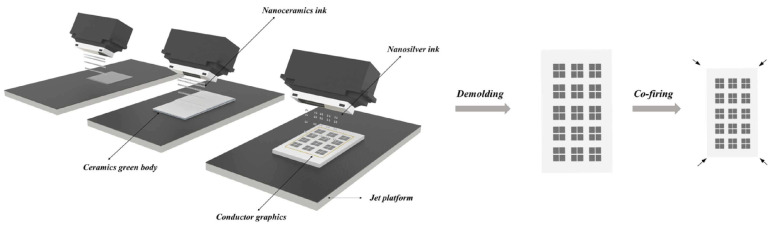
LTCC substrate and surface metallization by MJ technique [80].

**Figure 11 micromachines-15-01244-f011:**
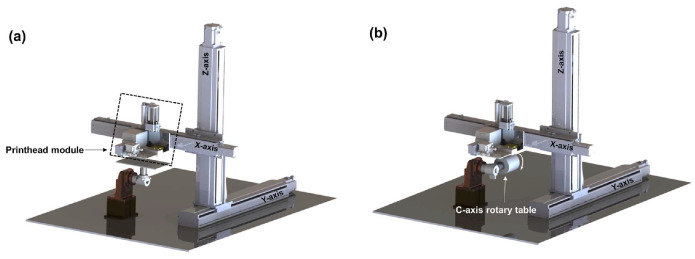
Machine for (**a**) flat and (**b**) curve printing of LTCC [80].

**Figure 12 micromachines-15-01244-f012:**
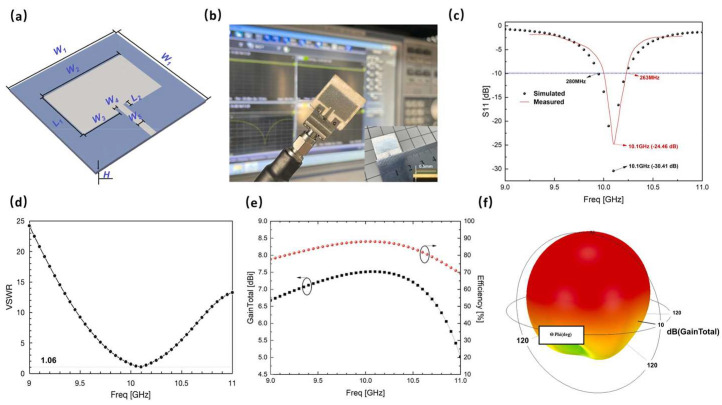
Microstrip patch antenna and RF measurements including S11, VSWR, and gain. (**a**) 3D perspective of the circuit (**b**) fabricated circuit (**c**) S11 simulation and measurement (**d**) VSWR simulation (**e**) gain and efficiency (**f**) 3D radiation [80].

**Figure 13 micromachines-15-01244-f013:**
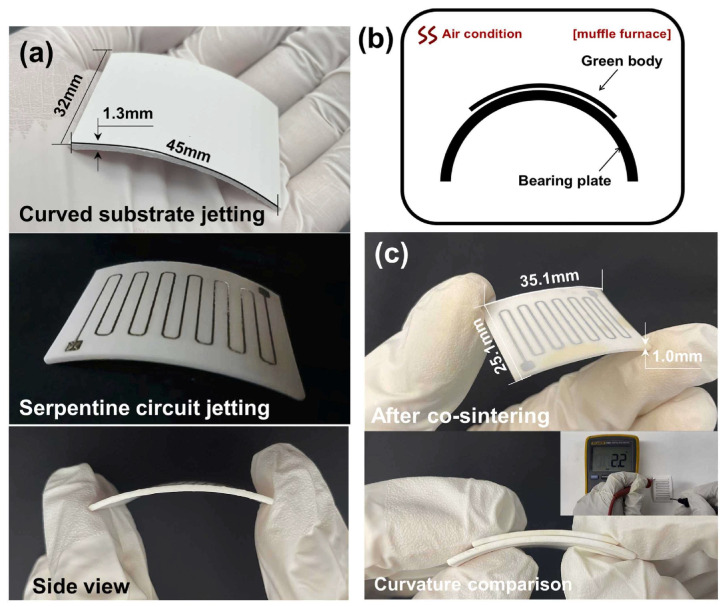
Fabricated curve LTCC with metallization on top by MJ technique. (**a**) fabricated curved LTCC (**b**) schematic diagram of the curved surface (**c**) shrinkage circuit after sintering with side views [80].

**Figure 14 micromachines-15-01244-f014:**
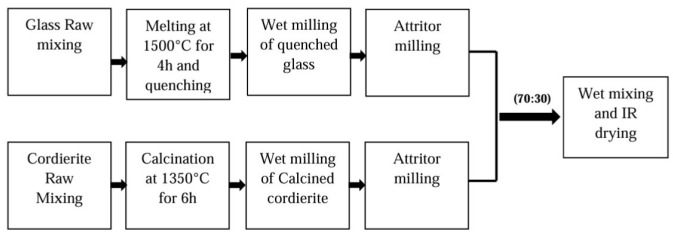
LTCC powder-preparation steps [81].

**Figure 15 micromachines-15-01244-f015:**
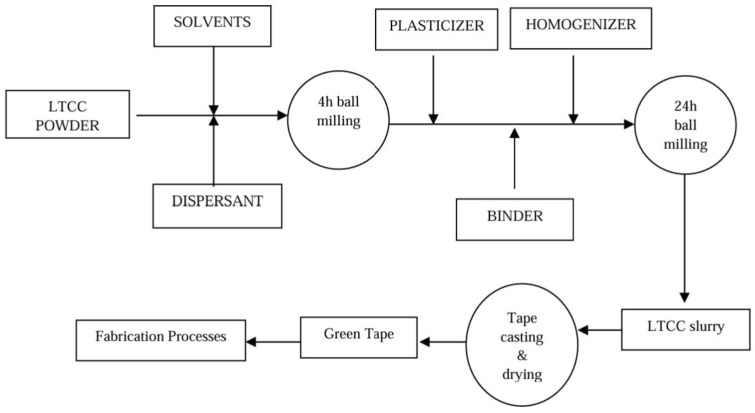
LTCC slurry and tape preparation [81].

**Figure 16 micromachines-15-01244-f016:**
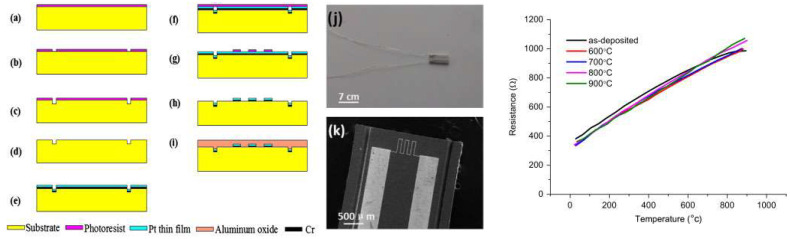
(**a**–**i**) Fabrication process of Pt film, (**j**) overall view of the sensor, (**k**) zoomed view of the sensitive area. Performance of the sensor on the right (resistance variation vs. temperature) [91].

**Figure 17 micromachines-15-01244-f017:**
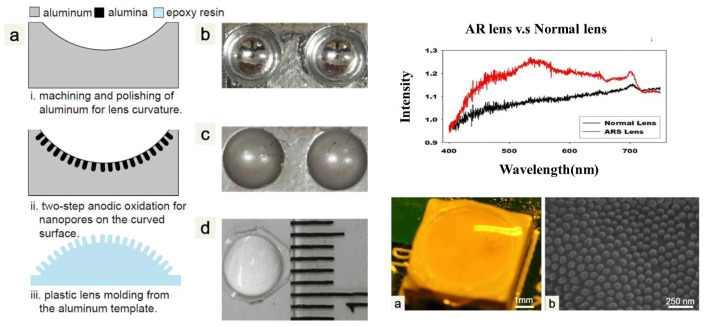
Fabrication process of AR lens (**a**) fabrication steps of AR lens, (**b**) polished surface of the curved aluminum (**c**) nanoporous alumina on curved aluminum (**d**) final optical image of the lens (**left**), AR lens vs. normal one comparison (**top right**), (**a**) AR lens on a yellow light, (**b**) nanopillars created by anodization of aluminum (**bottom right**) [95].

**Figure 18 micromachines-15-01244-f018:**
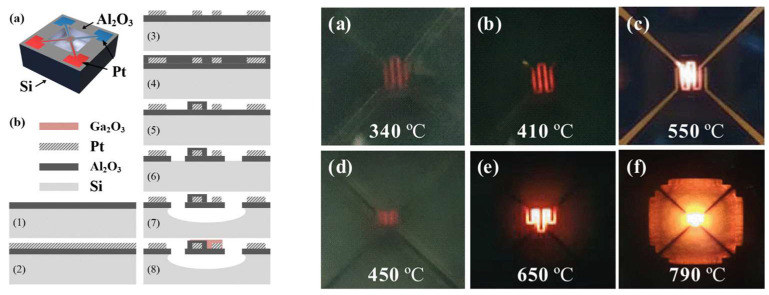
Alumina membrane gas sensor fabrication process (**left**), gas sensor under test in different temperatures (**a**–**c**) Si_3_N_4_ and (**d**–**f**) Al_2_O_3_ μHP (**right**) [98].

**Figure 19 micromachines-15-01244-f019:**
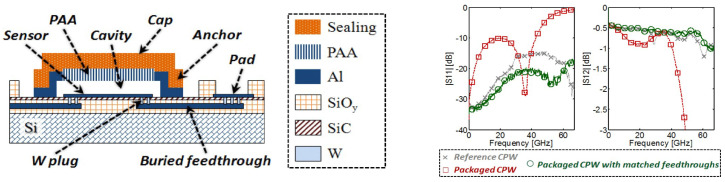
Fabricated bridge sealed with Alumina and silicon nitride (**left**). Measured S-parameters (**right**) [99].

**Figure 20 micromachines-15-01244-f020:**
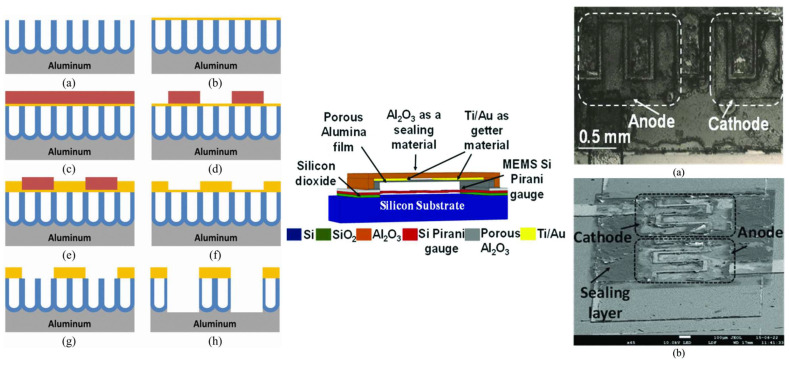
Fabricated alumina nanopores with high aspect ratio (**a**) Two-step anodization process. (**b**) Cu seed layer deposition process. (**c**) Photoresist spin coating process. (**d**) Photolithography and patterning processes. (**e**) Cu electroplating process. (**f**) Removal of photoresist. (**g**) Etching of cu seed layer. (**h**) Etching of AAO membrane (**left**). Thin film packaging using glow discharge (**center**) and fabricated view from top illustrating anode and cathode metals (**a**) top view (**b**) SEM image (**right**) [100,101].

**Figure 21 micromachines-15-01244-f021:**
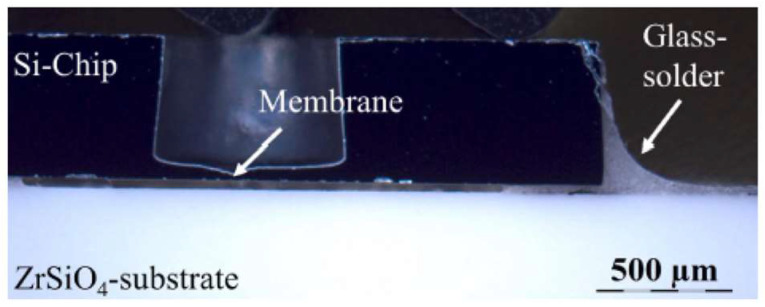
Flip-chip assembly on zirconia-silicate [103].

**Figure 22 micromachines-15-01244-f022:**
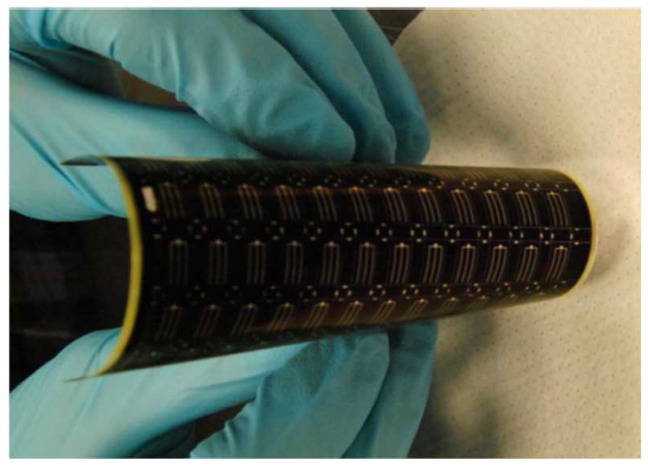
Fabricated flexible solar cell [104].

**Figure 23 micromachines-15-01244-f023:**
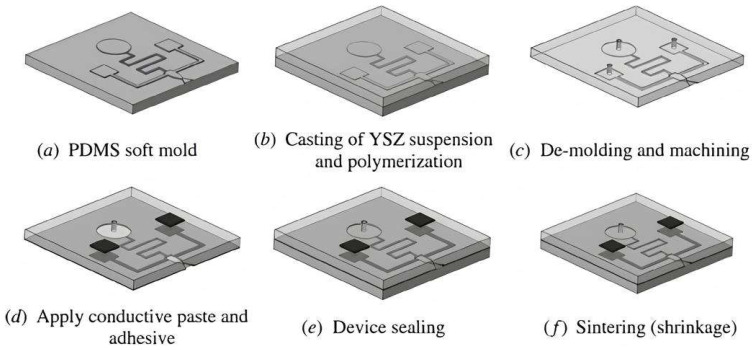
Fabrication process of micro-thruster [105].

**Figure 24 micromachines-15-01244-f024:**
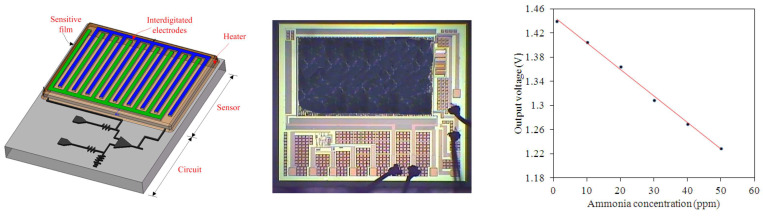
Ammonia sensor and readout circuit (**left**), fabricated circuit (**center**), measured results of the sensor (**right**) [106].

**Figure 25 micromachines-15-01244-f025:**
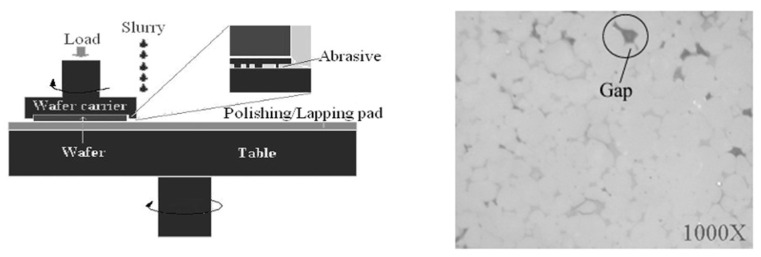
Polishing machine (**left**), gaps found on the surface of AlN after polishing (**right**) [108].

**Figure 26 micromachines-15-01244-f026:**
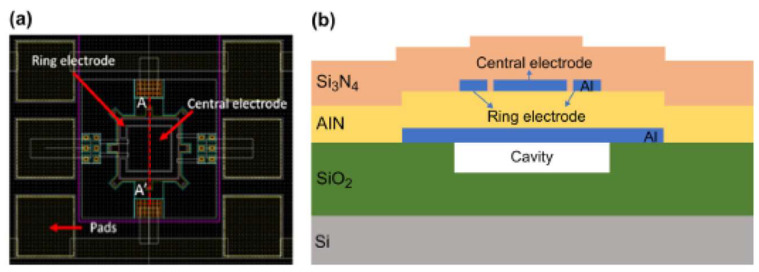
(**a**)A PMUT device top view (**b**) cross section with different layers including AlN piezo layer [112].

**Figure 27 micromachines-15-01244-f027:**
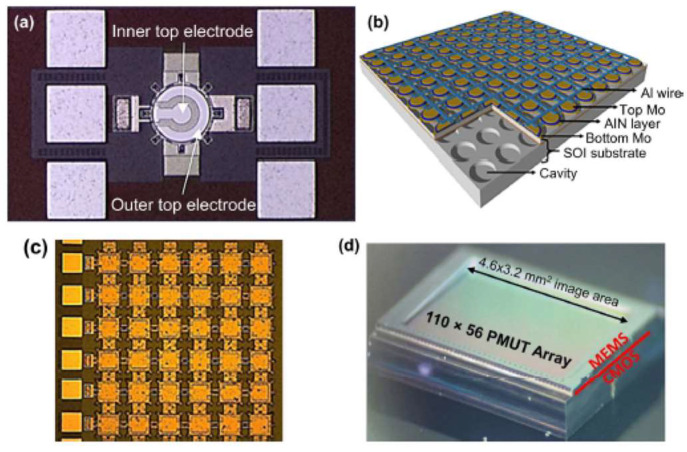
Different type of PMUT devices in arrays, (**a**) top view of a PMUT device, (**b**) arrays of PMUTs 3Ddesign, (**c**) top view of arrays of PMUTs, (**d**) dimensions of PMUT arrays as a MEMS chip on CMOS device [112].

**Figure 28 micromachines-15-01244-f028:**
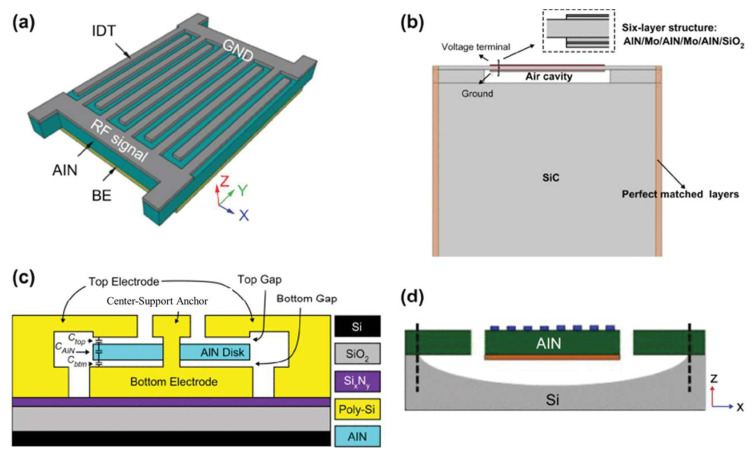
(**a**) A 3D view of AlN lamb wave resonator, (**b**) cross-section of AlN BAW resonator, (**c**) cross section of resonator with centered anchor, (**d**) cross section of a conventional lamb wave resonator [112].

**Figure 29 micromachines-15-01244-f029:**
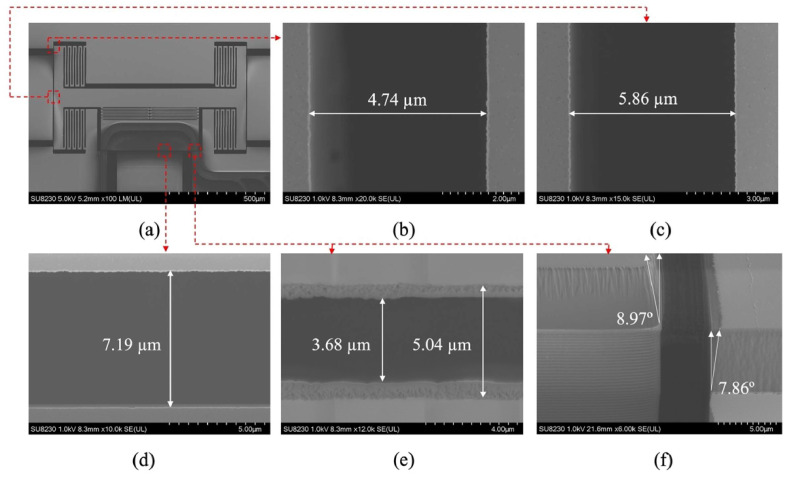
SEM image of the fabricated optical 3 × 1 switch with zoom views and dimensions, (**a**) fabricated device top view; (**b**) mechanical stopper gap; (**c**) switching actuator gap; (**d**) air gap of the gap closing actuator; (**e**) air gap closing interface; and (**f**) etch profile of the optical stack. [129].

**Figure 30 micromachines-15-01244-f030:**
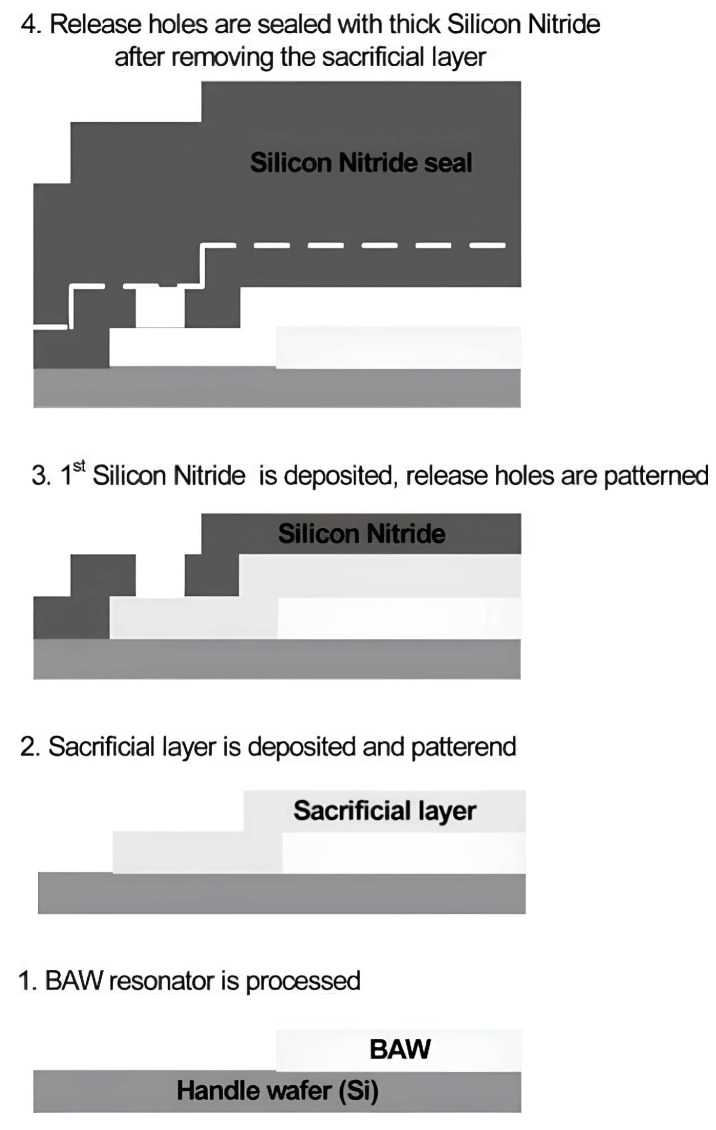
Silicon Nitride sealing fabrication process [130].

**Figure 31 micromachines-15-01244-f031:**
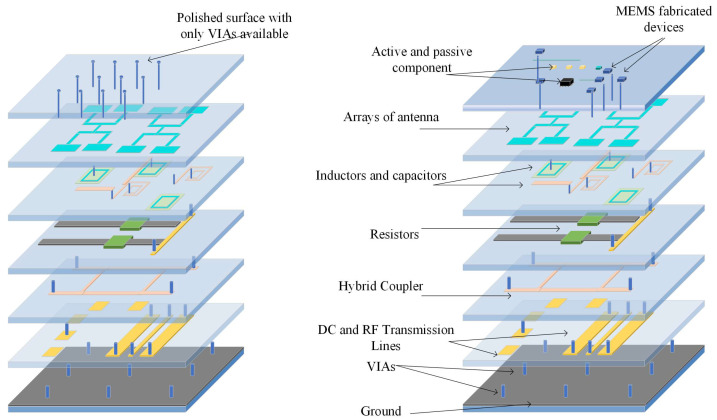
LTCC layers with embedded vias, cavities, and metallization as active substrate: (**left**) polished surface with vias on top, (**right**) active component and MEMS devices on top after final monolithic fabrication [134].

**Figure 32 micromachines-15-01244-f032:**
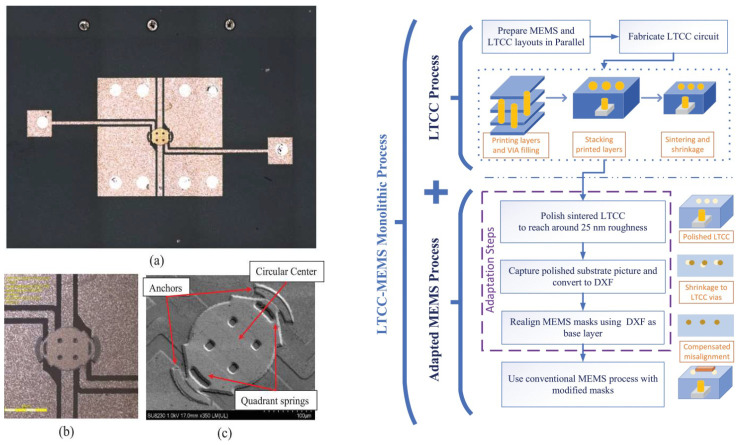
Fabricated capacitive MEMS switch with LTCC MEMS monolithic process, (**a**) Top image of the fabricated switch, (**b**) enlarged view (**c**) SEM image (**left**) LTCC-MEMS process flow (**right**) [134].

**Figure 33 micromachines-15-01244-f033:**
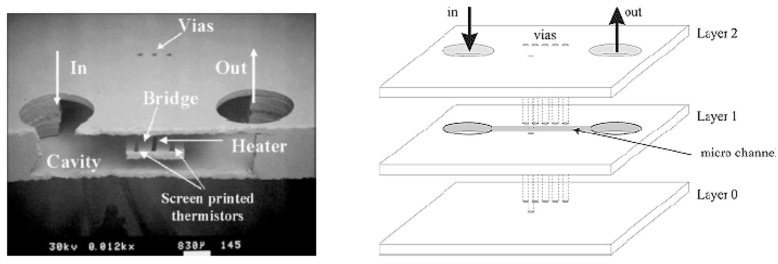
Cavities and via holes acting as a fluidic system for sensing application with embedded sensor [139].

**Figure 34 micromachines-15-01244-f034:**
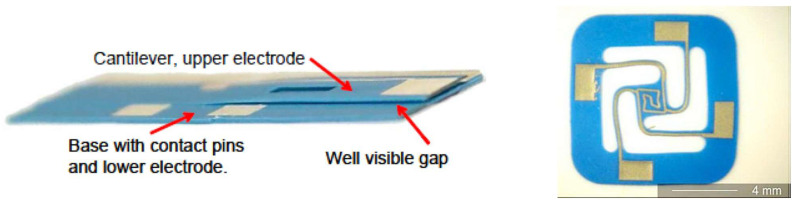
Fabricated cantilever with LTCC ceramic materials (**left**) LTCC hotplate (**right**) [139].

**Figure 35 micromachines-15-01244-f035:**
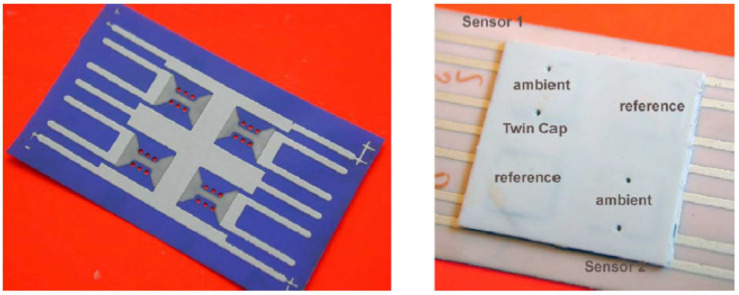
LTCC humidity sensors made of different LTCC ceramic materials [139].

**Table 1 micromachines-15-01244-t001:** Frequently used ceramic materials for MEMS and their properties [40,53,54].

Properties	Thermal Conductivity (W·m^−1^·K^−1^)	CTE	Dielectric Constant	Breakdown Voltage (volts/mil)	Flexural Strength (MPa)
Ceramic Materials					
Alumina	8–35	6.9–8	8.4–9.9	200–1090	120–400
Zirconia	3	10–11.4	28–33	100–150	710–1470
AlN	150	4.6–5.3	8.6	250–500	310
Si_3_N_4_	23–54	2.4–3.5	8.3–9.6	400–500	580–1020
LTCC	2–4.6	4.4–7	5.7–7.5	750–900	170–275

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
