# Peer review of "Ceramics for Microelectromechanical Systems Applications: A Review"

_micromachines, 2024, doi:10.3390/mi15101244_

Round 1
Reviewer 1 Report
Comments and Suggestions for Authors
Comments and revision are given in the attached report.

It requires an expert editor or copywriter to improve it.
Reviewer 2 Report
Comments and Suggestions for Authors
It is not clear what are the topics the authors want to include in the review. They mentioned in the review paper that - Alumina, zirconia, LTCC and SiN are the four ceramics which are used as the MEMS substrates. The
MEMS is a huge field (mechanical, optical, RF etc) and there are many MEMS devices in these areas. The authors perhaps can pick one or two fields and write the review based on that. The advantages and disadvantages of these substrates are not mentioned elaborately, nor they used a table to provide summary of the properties of the devices using Alumina, zirconia, LTCC and SiN as substrates. Below are the few other comments for the authors to consider as well:
1. In page three, table 1, the authors need to provide the references from which they obtained various data mentioned there.
2. Many of the subscripts such as TiO2, CaCO3 mentioned in page 5 is not written properly.
3. For Alumina, zirconia and SiN, add physical vapor deposition as one of the deposition methods.
Reviewer 3 Report
Comments and Suggestions for Authors This article presented a comprehensive review of the MEMS fabrication on Ceramics in different approaches. A very interesting topic and a very useful reference for the R&D in this field. This review is well organized and written, however, I still have the following concerns:- About the functional application in the field of AM of ceramics, maybe some related publications can help: Advanced Materials Technologies, 2024, 2401160. Journal of Materials Chemistry A, 2024, 12, 14479-4490.
- Please give the source for the data listed in Table 1.
- About the name of the AM of ceramics, the authors are suggested to modify according to the ISO standard or some published reviews/papers.
- For the AM of ceramics, how did you eliminate the cracks or defects among the materials? How did you characterize these. Some studies maybe helpful: Journal of the European Ceramic Society, 2023, 44(3): 1361-1384.
- Please pay attention to all your figures and tables, they are suggested to modify to meet the publication requirements.
- Some labels or words in the figures are not clear enough to read.
Round 2
Reviewer 2 Report
Comments and Suggestions for Authors
None
Reviewer 3 Report
Comments and Suggestions for Authors
The authors addressed all my concerns. The manuscript is recommended to be accepted as it is.